# A Community-Engaged Ethnographic Investigation into Public Transit Among Older Adults Experiencing Homelessness

**DOI:** 10.3390/ijerph22040654

**Published:** 2025-04-21

**Authors:** Whitney Thurman, Tara Hutson, Dylan Lowery, Amy Patten, Alexandra A. Garcia

**Affiliations:** 1School of Nursing, The University of Texas at Austin, Austin, TX 78712, USA; amypatten@my.utexas.edu (A.P.); agarcia@mail.nur.utexas.edu (A.A.G.); 2College of Pharmacy, The University of Texas at Austin, Austin, TX 78712, USA; t.hutson@utexas.edu; 3RBJ Senior Housing, Austin, TX 78702, USA; dlowery@rbjseniorhousing.org

**Keywords:** unhoused, aging, public transportation, qualitative, mobility

## Abstract

Nearly 50% of single homeless adults are over the age of 50, and adults aged 50 years and older are the fastest growing subpopulation of the homeless in the U.S. This subpopulation has unique mobility needs and barriers to transportation. We used rapid ethnographic assessments to explore the practices, needs, perceptions, and values of unhoused older adults in relation to their efforts to access and navigate health and social services via public transit. We conducted 23 observations of 12 participants as they navigated public transit to meet their travel needs. Our data consist of 65 h of observations with field notes, walking interviews, and travel diaries. Two themes—waiting and friction—encompassed participants’ experiences of mobility and their ongoing negotiations that involved time, space, individual ability, and interpersonal interactions within their environmental context. For this population, attainment of housing and improved health and well-being is contingent on access to reliable transportation—a condition that is not met in many communities. For the participants in our study, the physical demands of accessing public transit combined with the cognitive load of interpreting multi-step directions in specific time schedules were often insurmountable. The public transit system was often hostile, such that participants were stigmatized and segregated. Professionals who work with older homeless individuals must consider the capacity of their clients to successfully navigate public transit, and policymakers should consider the transit needs of their entire population when designing transit systems.

## 1. Introduction

Mobility is important for individuals’ healthy aging; it influences their health, well-being, social connection, and independence [1]. It consists of the ability to move oneself, with or without assistance, within one’s home, neighborhood, and beyond [2], and it is facilitated by modes of transportation such as cars, trains, and buses [3]. Optimal mobility consists of the ability to go safely and reliably where one wants to go, when one wants to do so, and by the means of one’s choice [4]. This capacity to meet the demands of the environment in order to move within and between environments is thus central to older adults’ quality of life. The significance of mobility for well-being in later life has been extensively studied within gerontology, transportation research, and sociology.

In 2023 in the U.S., over 650,000 individuals were estimated to experience homelessness on any given night [5], nearly half of whom were aged 50 years or older [6]. This population is increasing, and the accelerated aging of unhoused individuals in comparison with the general population is well-known, with unhoused adults considered to be “older” by age 50 [7,8]. Compared with the needs of younger unhoused populations, those of older adults experiencing homelessness (OAEH) are often more complex and include medical, social, and functional support needs [9], and most homeless adults aged 50 years or older have at least two chronic health conditions [10,11]. In a review of frailty among homeless adults, Mantell et al. (2023) synthesized evidence that frailty presents earlier and at higher rates in homeless populations than in community-dwelling cohorts even when they are compared with cohorts with very low incomes [8]. In a sample of 350 homeless individuals aged ≥ 50 years, Hurstak et al. (2017) found that 25.1% displayed impairment in global cognition and another 32.9% exhibited significant impairment in executive function [12]. These rates are 3–4 times higher than those of housed adults aged over 70 [12]. Thus, OAEH may experience mobility impairments due to age-related changes in balance, gait, and cognition [13] or to chronic illness, frailty, depression, lack of social support, or any combination of such conditions that disproportionately impact this population.

Mobility among OAEH is impacted by the need to rely on public transit to meet travel needs [14], because public transit is the most common means of transportation among the unhoused [15]. In the U.S., this reliance on public transit is in stark contrast to the practices of the housed population, who rely predominantly on personal vehicles for which our communities have been designed and built [16]. As a result, the majority of research on aging and transportation has focused on older adults with personal vehicles, although more recently, attention has turned to public transit use among older adults [17,18]. Public transit can extend mobility to individuals who are disadvantaged with respect to transportation—those who lack access to a personal vehicle, who are too young or old to drive, or who live with disabilities that prevent their driving. Yet barriers such as affordability, physical inaccessibility, or limited services often undermine the ability of public transit to serve as an effective substitute for personal vehicles [19,20], and these barriers contribute to unmet travel needs—“mobility needs that remain unfulfilled due to the inability to accomplish needed or desired journeys and activities” [21].

Among older adults, travel needs encompass practical activities such as attending medical appointments or grocery shopping, psychological and physical benefits of movement, and expressions of personal autonomy and freedom [22,23,24,25]. Research has identified that these needs are unmet among at least one third of older adults, and poor health is the most common contributing factor [21]. Poor health impacts mobility by reducing one’s range of activities, owing to the need to prioritize and conserve energy; and in people who cannot drive, physical limitations can interfere with using public transit because they must walk to and from bus stops as well as get on and off the bus [26].

Within the general unhoused population, evidence clearly indicates that transportation disadvantage impedes access to community resources needed to regain stability, thereby creating barriers to housing, social support networks, employment, and health promotion [14,27,28,29]. However, much of the research on homelessness and transportation has relied on surveys or on retrospective questionnaires to elicit participants’ experiences [29]. Although recent research has explored the perceptions of homeless services providers regarding how mobility impacts service use, delivery, and engagement [27], this may not fully capture mobility barriers or transportation disadvantage [30,31]. Little to no research has specifically explored mobility and travel needs among OAEH. It is likely that in this distinct subpopulation of aging adults who lack safe and stable housing and live with high levels of medical complexity, travel needs and motivation for mobility may be different than they are for older adults who are housed or for younger unhoused individuals. Current evidence suggests a critical need to better understand how OAEH meet their travel needs while being reliant on public transit. In this study, therefore, we explore the experiences, practices, perceptions, and values of OAEH in relation to their efforts to access and navigate public transit.

### Sensitizing Framework

This study is informed by Luiu et al.’s (2018) conceptual framework for assessing unmet travel needs [32]. Included concepts are travel patterns and access to transport modes, attitudes toward transportation, coping, and planning. We situate these concepts within Cresswell’s (2010) argument for a politics of mobility in order to facilitate our examination of how OAEH access and navigate public transit to meet their travel needs and how mobility may be implicated in the ongoing marginalization and stigmatization of OAEH [3]. Together, these concepts guided our data collection and served as a sensitizing framework for the interpretation and analysis of the data [33,34].

## 2. Materials and Methods

Guided by principles of community-based participatory research [35], this study reflects the efforts of a workgroup within the local homelessness response system that was tasked with investigating potential solutions to transportation barriers faced by unhoused individuals. The workgroup includes direct service providers, policymakers, researchers, and people with lived experience of homelessness. Both academic (XX) and community partner (XX) authors of this study are members of the workgroup. This study extended the efforts of the workgroup and aimed to understand the experiences of mobility and public transit among OAEH. Subsequently, both academic and community partner representatives were engaged in every stage of the research.

In conducting this research, we employed rapid ethnography. Traditional ethnography seeks to experience and understand the world through the eyes of others by actively participating in their daily lives [36]. Rapid ethnography is a specialized ethnographic methodology that prioritizes efficiency and timeliness without sacrificing depth of understanding [37], and it is well suited to producing actionable findings that can inform policy and practice [36]. Rapid ethnographic methods do not pose an undue burden on study participants and are therefore both methodologically and ethically appropriate for gaining a contextual understanding of how OAEH navigate their environments. Unstructured walking interviews, observation of participants, and travel diaries provided a detailed understanding of the daily experiences, routines, feelings, and frustrations of participants as they navigated public transportation and interacted with others in public spaces. Ethical approval was obtained from the University Institutional Review Board.

### 2.1. Study Setting

This study was conducted in Austin, TX, USA, where homelessness is a persistent concern and the number of older adults experiencing homelessness is steadily increasing [38]. Like other communities, the homelessness response system in Austin is complex and includes an array of entities providing social services, healthcare, and housing supports. Homeless services are rarely co-located, requiring PEH to navigate across systems and geographic distance to access needed resources and support.

### 2.2. Recruitment

Recruitment relied primarily on referrals from an emergency shelter where our community partner agency had offices. Case managers received a presentation about the study, eligibility criteria, and incentives for participation, and the research team provided recruitment materials to facilitate conversations with clients. Case managers referred clients who met inclusion criteria and expressed interest in participation to the research team, who then followed up directly with each client to schedule a time for study enrollment. The research team also worked with staff at the shelter and at a local navigation center to identify potential research participants, where the referral process was the same.

### 2.3. Participants

Individuals were eligible for participation if they were currently experiencing homelessness or currently housed through a rapid rehousing program, enrolled in case management with our community partner, and at least 50 years old. Individuals with the inability to provide informed consent or to communicate verbally in English were excluded. All participants provided informed consent as well as permission to be audio recorded at the end of the study during an exit interview.

### 2.4. Data Collection

After approval by the University Institutional Review Board (STUDY00003005), data were collected by the first two authors from September 2022 to May 2023. Walking interviews and observations of participants were conducted twice during a 3-week time period for each participant. To explore participants’ range of travel needs (practical activities, psychological and physical benefits of movement, and expressions of personal autonomy and freedom), we accompanied each participant once on a day when the participant had a scheduled appointment and once on a day without an appointment. Participants also received small notebooks and pens and were asked to record a travel diary 3 days per week during the 3-week study period. We asked them to record reasons for traveling, notable events while traveling, and whether they successfully met their travel needs on their particular travel trips. At the end of the second observation day, participants returned their travel diaries and participated in exit interviews. Participants received $25 upon study enrollment, $10 after the first observation day, and $25 at study exit. Figure 1 depicts data collection strategies and timepoints.

Twenty-three field observations totaling 65 h were conducted. Transit observations ranged from 45 min to 6 h, during which the researchers engaged in conversation and unstructured interviews with the participants. Field notes documented during and immediately following the transit trips and the walking interviews focused on participants, settings, and destinations. We documented participants’ characteristics including functional ability; where, when, why, and how participants traveled; participants’ perceptions of the travel experience; interactions with others and with systems; unanticipated events and reactions to those events; and settings, including environmental conditions. Exit interviews ranged from 15 to 45 min and focused on participants’ experiences of the study and their perceptions of environmental modifications that might facilitate their ability to travel to important destinations.

To characterize and describe the sample, at enrollment, participants completed a series of surveys regarding sociodemographic and health-related characteristics. To facilitate the ability of participants with low literacy levels or low vision to complete the surveys, the first or second author administered the surveys by reading the survey items out loud. The response options were also read out loud and provided in written format so that participants could respond verbally or point to their response choice. Sociodemographic characteristics included gender, race, employment status, veteran status, and income. Functional ability was assessed using the PROMIS Physical Function with Mobility Aid short form [39]. This is a brief measure of an individual’s self-reported ability to stand and move with and without support. It generates a t-score that is compared against the general population’s mean score of 50. We assessed transport difficulty using 15 items that ask participants to rate the level of difficulty they experience with various aspects of transportation, including travel costs, time, accessibility, availability, safety, and awareness. Response options ranged from 1 = very easy to 5 = very difficult. Total scores ranged from 15 to 75 with higher scores indicating higher transportation difficulties.

### 2.5. Data Analysis

Exit interviews were de-identified and professionally transcribed. Handwritten travel diaries were transcribed by a graduate research assistant. After verifying the accuracy of the transcriptions, we entered the interview transcripts, travel diaries, and field notes into NVivo software version 14 for qualitative data management. Descriptive statistics were used to analyze the quantitative data and were calculated using Microsoft Excel Version 16.96.

We used thematic analysis with interrelated inductive and theoretical components [40]. The interview transcripts, field notes, and travel diaries were reviewed throughout data collection to clarify any points of confusion. For example, if there was uncertainty regarding the meaning of something that a participant said during an interview or recorded in a travel diary, we consulted the field notes for additional context regarding the participant’s observed transit experiences. In this way, multiple sources of data facilitated triangulation. Coding was iterative, aligned with Braun and Clarke’s (2006) six-phase approach: (1) familiarization with the data, (2) generation of initial codes, (3) search for themes that cut across responses, (4) review to identify main themes and subthemes, (5) definition and naming of themes, and (6) a final report [40]. We first read all transcripts, field notes, and travel diaries to ensure familiarity. We then collaboratively segmented and labeled two interview transcripts and two sets of field notes to generate an initial list of codes. Patterns of meaning in the data were examined and discussed during weekly research team meetings, resulting in an initial code book for OAEH’s experiences of mobility. Next, we identified themes based on explicit meanings in the data. For example, data coded as “figuring out where/when to go” and as “technology use” were collaboratively reviewed and ultimately collapsed into the category “wayfinding”. We then compared these emerging themes with our sensitizing framework as an analytical tool and reviewed the coding structure to refine and organize a final set of themes, which were agreed upon by the project team. Our final step was to review the completed coding to reach consensus on thematic saturation [41].

### 2.6. Rigor

We used several strategies to ensure rigor and to strengthen internal validity. First, a five-person research team that included a community partner who was a direct service provider analyzed the data to enhance credibility and minimize bias. Throughout the study, ongoing reflexive dialogue between the researchers and community partners facilitated attention to assumptions that often stemmed from our own social positions as housed community members who rely on personal vehicles for the majority of our transportation needs. Other strategies included reflexive journaling, consideration of negative cases, and maintenance of an audit trail using NVivo Version 14. Finally, rich description of the study context, participants, and methods allows readers to gauge the transferability of the findings to other communities [42].

## 3. Results

Twelve individuals enrolled in the study (Table 1). Their mean age was 54.8 years (range = 50–59); four were women. Seven were staying in a bridge shelter as they awaited housing, two were unsheltered, two had been re-housed, and one was temporarily in a motel. One participant was unable to complete the transit trips and exited after the first observation day; her data are included in the results that follow. Observed transit activities and intended destinations are listed in Table 2.

Qualitative analysis identified two themes—waiting and friction—that encompass participants’ experiences of mobility and their ongoing negotiations of time, space, individual ability, and interpersonal interactions within their environmental context. Participants navigated the unreliable and sometimes hostile public transit system in pursuit of resources, information, and social connection as well as for something to do and to stay hidden from stigma, criminalization, and ever-present threats to physical and psychological safety. In doing so, participants encountered people, places, and experiences that alternately served as motivators and deterrents to their continued efforts to meet their travel needs.

### 3.1. Waiting

This theme encompassed the time burden associated with homelessness and with the forced dependence on public services including public transit. Categories included *everyday mobility* and *overall experience of homelessness*.

Everyday mobility. Waiting, whether for the bus, in line for food, to speak to a case manager, for one’s friend to arrive, or for any number of other mundane needs, composed participants’ rhythms of daily life and dictated their mobility. It was rare that any participant completed more than one task in a given day, because of the time they spent waiting for various things. Being dependent on public transportation meant that waiting for the bus was a regular part of everyday mobility. Buses usually came at regular intervals, but depending on the time of day and day of the week, participants could be left stranded for an hour or even overnight. Participants had learned that they could not trust this system, so they generally did not expect to arrive anywhere on time, even if they planned ahead:


*Don’t push it because the buses are always late when you’ve got an appointment…It’s a waiting game. You wait, wait, wait, wait…but once you learn the system and you get used to it, it’s just part of the package. It’s part of the ride.*
(Exit Interview, 9/14/22)

Because of this inability to control their time, participants tried to avoid scheduling appointments that required strict arrival times. On occasions when participants did need to be punctual, they would depart hours before the scheduled time in anticipation of delays and confusion. This strategy precluded participants’ ability to accomplish anything else on a given day, and the trip would be colored by nervous energy because participants knew that missing the appointment would mean having to restart the entire scheduling and traveling process again. One participant recounted missing a scheduled visitation appointment with her daughter, who was in the county jail, due to missing a bus connection and was noticeably anxious about avoiding that same experience on her visit scheduled for the following day:


*How many hours do I got to get up earlier just to wait for the 271?*
(Exit Interview, 10/3/22)

Depending on the participant’s destination, arrival often restarted the process of waiting. Lines for food stretched down sidewalks, and lines for services and resources—to obtain clothes and hygiene supplies, collect mail, schedule appointments—were ubiquitous:


*It takes [participant] 45 min to reach the front of the line. I am not sure what he requested at the table, but it appears he waited that long to get a pair of shoelaces and a sweater.*
(Field Note, 3/21/23)

Despite knowing that they would have to wait, participants sometimes opted to physically travel to a specific location to try their luck at walking in for same day appointments, or, failing that, to schedule an appointment for a future date instead of calling and being “put on hold forever” (Field Note, 10/12/22). Traveling to schedule future appointments also served as motivation for getting up and about.

In addition to the daily rituals of queuing, participants’ lives were essentially on hold as they waited for housing, for a job interview, for disability benefits, and for any other resource that required interaction with the bureaucratic homeless services system. Active efforts toward stability on the part of the individual were punctuated by interludes of frustrated waiting and required constant negotiation between traveling to meet short-term needs for survival and remaining in place for the possibility of securing longer term resources for stability:


*He tells me he has a job interview this week and has a few other appointments, so he has less time to be waiting in lines. He tells me about trying to get his ID renewed and how he would take the bus to go to the only location that takes walk-ins. He said he would get there at 7:30 a.m. and wait all day just in case there was a chance he could be seen. He said if you were lucky someone would come out around 2:30 or 3 and tell you there’s no way you’ll be seen. If not, then he would wait until 5 p.m. He said the hard part about that is that means you then have to forego eating. He said he did that for 4 days in a row before he finally gave up. He made an appointment, and it was a 5-month wait.*
(Field Note, 4/5/23)

Relying on others’ assistance in navigating siloed systems did very little to alleviate waiting and sometimes added significant burden to the process. One attempt to attend an appointment scheduled by a case manager resulted in an unsuccessful 4-h trek across town from the shelter to an optometrist and back. The physical inaccessibility of the location, combined with confusion about to how to get there, resulted in walking over 4 miles in 90° weather, only to be rescheduled for a new appointment 3 days in the future (Field Note, 9/16/22). At other times, the system itself perpetuated waiting:


*It is a little after 11 a.m. at this point, and they close at 1 p.m. At first, it doesn’t seem like the line is moving very quickly but after about 20 min it starts to move a little faster. We are probably about 7 people from the front…The time is getting closer to 1 p.m. now and the line for food has shortened….a few minutes later…[participant] said they didn’t have what she needed but was told to come back on Wednesday to appeal her disability and on Thursday to get her prescriptions.*
(Field Note, 10/3/22)

### 3.2. Friction

To the extent that *waiting* constituted participants’ everyday mobility, *friction* helped to define it. The theme *friction*, whether as resistance or as the process of stopping itself, was inherent in the unhoused older adults’ navigation through the world. Often, waiting *caused* friction, contributing to a negative feedback loop of missed opportunities and ever greater inertia. Categories include *wayfinding, interpersonal interactions,* and *accessibility.*

*Wayfinding.* Being reliant on public transportation to meet travel needs necessitated regular, if not daily, bus trips. Despite this, participants encountered resistance when attempting to figure out where and when to go in order to meet those needs. Limited access to technology and electricity to keep devices charged, as well as lack of digital literacy among those who did have smartphones, regularly interfered with wayfinding. Out-of-date websites undermined participants’ ability to know when and how to access the bus to reach specific destinations for specific purposes. In response, several participants established travel routines using specific routes and tried not to deviate from these known paths:


*[Participant] says he really only rides this route, but it is great because it takes him to the places he needs to go.*
(Field Note, 3/21/23)

Some participants were reliant on the bus drivers to alert them to their stops; when drivers failed to do so, we would miss our exit. This would result in extra time on foot, backtracking to the intended destination. At other times, participants responded by abandoning their original destination. Plans were also changed depending on which bus arrived first:


*About 15 min pass, and the next bus comes. It is another #7 bus (the 310 has not yet passed). He said, “I guess we’re going downtown.” We had set out with a poorly defined plan to go to [clinic] so [participant] could see about getting an appointment to get back on his psych meds. As we were walking towards the bus stop, he muttered a little bit to himself about where [clinic] actually is. He mentioned that someone had told him there was one that was not too far away. Because he doesn’t have a phone, he couldn’t look for an address or directions. But it still seemed as though we would try. However, when we got on the #7, [participant] said that we would go downtown and get some job applications.*
(Field Note, 3/9/23)

Participants also faced resistance from their own changing health status and ability. One was not physically capable of walking to the nearest bus stop from her shelter, nor was she able to process written directions from the shelter staff to find her way there. Others had the physical capacity to do so, but cognitive function interfered with their wayfinding. When asked how he figured out how to get to where he needed to go, one participant said,


*Trial and error. Right after I had the stroke, there was a lot of error…I would get on the wrong bus.*
(Exit Interview, 4/11/23)

Most others lived with chronic pain, and some were adjusting to being reliant on public transportation because they were no longer physically capable of riding their bicycle, which had been their preferred method of transport.

*Interpersonal interactions.* Interpersonal interactions were perhaps the greatest source of friction in participants’ day-to-day lives, as participants relied on the goodwill of others to successfully navigate in the community. The action—and inaction—of individuals within various settings could be the difference between a kept appointment and a wasted effort. More than once, bus drivers simply did not stop to let us on the bus. Some attributed drivers’ actions to discrimination based on their appearance or their neighborhood, and others knew that if they did not wait directly in front of the bus stop—even if it was in the direct sun—the driver was unlikely to stop. The driver’s actions did not have to be malicious for the impact to reverberate throughout the participant’s day. At times we waited for the next bus following a missed connection, but more commonly participants decided to forego the planned activity, and we would either return to the shelter or completely alter course. Participants were also greeted with hostility from other bus passengers:


*When I get on the bus, I ask to sit in the front seats…I have problems with my legs. She told me to go sit somewhere else.*
(Participant Travel Diary, 10/20/22)

*Accessibility.* Navigating public transit and public systems was fraught with access barriers and, at times, was rendered completely inaccessible by functional ability, environmental conditions, public designs, or personal lack of knowledge about resources. Most participants had an impaired gait and recounted a litany of conditions or past injuries contributing to ongoing pain and mobility impairments. However, none regularly used an assistive device such as a cane or walker, contributing to the ever-present threat of injury from falling: [participant] states her left knee is really hurting her because she fell. She shows me where it is visibly swollen on the side and behind her knee (Field Note, 10/14/22). Participants perceived the bus to be precarious both for themselves and for others with limited mobility, and they were aware that falling was a significant hazard:


*A person without good balance could get injured real bad…If I fall and break something, it’s a big thing, because…I’m old. You know what I’m saying?*
(Exit Interview, 3/9/23)

Accessibility was also impeded due to long and difficult walks to the bus stop. Within one half-mile of the shelter where most participants stayed, there were two bus stops. One was uphill with a wide sidewalk but no trees, along the frontage road of a major interstate. The other was slightly downhill through a neighborhood with several trees providing shade but no sidewalks. Neither of these was easily accessible for people with chronic pain or limited mobility, which created resistance before participants ever departed from the shelter:


*This time there is no sidewalk, and we are forced to walk in the uphill slanted grass along the road. The road does not have a shoulder either, so traffic passes us very close by. The terrain we are walking through is very uneven and [participant] comments that he didn’t know we would be hiking today.*
(Field Note, 9/16/22)

Bus stops did not reliably offer shelter from the elements, and participants were keenly aware of which ones offered shade, benches, or trash receptacles. They regularly pointed out bus stops that lacked these amenities, questioning why some had been deemed worthy of providing such comfort whereas others had not, and how those decisions were made. Regardless of the walking and waiting, a climate-controlled bus served as temporary respite from harsh environments. However, stepping foot onto the bus was sometimes tense as participants negotiated access because they did not have bus fare or, more commonly, struggled with the fare machine. Bus fare could be paid with dollars, coins, credit cards, bus passes, or a mobile app, and the machine required specific money or card orientations. Participants were rarely able to insert their bus pass or fare correctly on the first or even second try, which served as a source of embarrassment and frustration, because bus drivers did not typically have much patience for such failed attempts:


*The bus driver when I got on, I put my ticket in backward and upside down two times. She just looked at me like, “figure it out?”*
(Travel Diary, 3/12/2023)

Services and resources were sometimes inaccessible because participants were unaware of their existence. This was true even for programs that were designed specifically for their benefit; none of the participants were aware of a temporary free bus fare program that had been instituted in the local community in response to extreme heat.

## 4. Discussion

In this study, we sought to explore the experiences, practices, perceptions, and values of OAEH in relation to their efforts to access and navigate public transit, and the results contribute to a growing body of literature regarding older adult homelessness. The quantitative data clearly indicate that participants had a high burden of chronic illness, physical limitations, and difficulties with transport. These findings are in line with prior evidence documenting the poor health of OAEH. Additionally, while the salience of transportation barriers in the everyday lives of unhoused individuals has been documented [27,43,44] the qualitative findings from the current study provide a more in-depth understanding of the mobility experiences of OAEH.

Participants in our study reported some difficulty in covering the costs of their transportation and some had to hustle for a bus pass or gain favor with the bus driver to be allowed on. Commonly, however, participants were provided bus passes by case managers or shelter staff, overcoming the affordability barrier. Participants nevertheless struggled to meet their travel needs, and many of our planned transit trips were ultimately unsuccessful. These findings contrast with earlier evidence suggesting that access to public transportation is associated with fewer missed healthcare appointments among people with low incomes [45], indicating that mobility among OAEH may be distinct from other people with low income. Our findings do, however, support prior evidence documenting that affordability is necessary yet insufficient to ensure accessibility, as research has indicated that a free fare policy does not necessarily result in increases in public transit use [46]. Similarly, increasing affordability alone has not been found to increase patient uptake of transportation services and has had an equivocal impact on health utilization or outcomes [47].

Our participants used several strategies to make and attend appointments, but their efforts were often undermined by an unreliable transit system. The temporal burden associated with public transit was at times not worth the hassle of attempting to attend appointments or meet other travel needs; as Cresswell (2010) has claimed, being able to get somewhere quickly may be a privilege reserved for a select few [3]. The temporal burden associated with mobility among OAEH permeated participants’ lives and was similar to that experienced by people with disabilities, in that they had a relatively constrained range within which they could participate in activities and were forced to spend much more time in accessing those activities [48]. Thus, the mismatch created by being reliant on the unpredictable public transit system and our cultural expectations and requirements of punctuality undermines any effort of OAEH to attend appointments, adhere to medical advice, and complete tasks needed to attain housing or other benefits. Urban planners and those tasked with the development and management of public transit systems should focus efforts on increasing the reliability of mass transit. Adding or upgrading dedicated bus lanes, optimizing traffic signals to reduce congestion, and increasing the frequency of transit trips are evidence-based strategies that could be implemented in local communities to make public transit more reliable [49].

It is also worth noting that despite impaired gait, fall history, and chronic pain that caused substantial mobility limitations, none of our participants used assistive devices. Our study did not specifically explore the use of walkers or canes for mobility, but prior research has documented difficulties associated with appropriate use of assistive devices among homeless populations. These difficulties stem from loss and theft [50] and from encampment sweeps resulting from ongoing efforts at criminalizing homelessness [51]. Other barriers related to the use of mobility devices include financial barriers and inadequate insurance coverage. Future research should further explore the use of assistive devices among OAEH, including facilitators and barriers to use and how they impact mobility and transportation.

Taken together, our findings have implications for health and social services professionals who work with OAEH as well as for policymakers and urban planners tasked with transit design. OAEH often experience impaired executive functioning and may lack practical skills such as route planning and an understanding of timetables that are needed for effective use of public transit [52]. Individuals in this population are also often disconnected from reliable sources of information, indicating a need for education about community resources as well as education that targets their navigation of public transit and social service systems [44]. Thus, professionals who work with this population should consider their clients’ capacity to successfully navigate public transit, and support may need to include more than the provision of bus fare. For example, case managers should consider transit accessibility, including bus stops and number of transfers, when scheduling clients for different health and social services. Service providers could also consider client transportation barriers when establishing policies related to late arrivals and no-show appointments. The cognitive burden of navigating the public transit system could be alleviated by the implementation of a peer support program, which has been shown to be useful in helping PEH with life skills among many other health-related benefits [53]. Transit systems should also ensure that their websites and bus routes are kept up-to-date, and entities that work with homeless populations should include bus route information on their websites and printed informational materials to improve accessibility.

### Limitations

In this study, we have investigated a small group of OAEH and their experiences of mobility in navigating public transit within a large metropolitan area of the south-central U.S. This small sample and setting are not generalizable to other OAEH in other geographical locations. However, generalizability is not the goal of qualitative research. Instead, transferability seeks to aid the reader in understanding how they can connect elements of this study to their own context and experience. In presenting this research, we have addressed transferability by providing detailed descriptions of the participants, data, and context, with direct quotations. The study’s ethnographic assessments provided a richer perspective of participants’ attitudes, feelings, and knowledge about their community and offered a fuller understanding of their spatial practices, which aids transferability and enhances trustworthiness. However, it is possible that the presence of the researcher during transit trips influenced participants’ behavior. To minimize this, we sought to build rapport with participants by conducting longer observation sessions and having the same researcher conduct both observations with each participant. We also had participants record travel diaries on days in which the researcher was not present to capture participants’ experiences and thoughts in real time. Finally, we did not verify our interpretations with the participants, which might also be seen as a limitation.

Future research should investigate mobility and transportation among rural OAEH, because rural infrastructure is notably insufficient for individuals who are disadvantaged with respect to transportation. In addition, OAEH are not a monolith, and so it is possible that despite the high burden of chronic illness experienced by the sample in the present study, their ability to participate successfully in the study may indicate that they constituted a higher functioning subset of OAEH. Future research could explore the impact of different diagnoses and functional abilities on participants’ experiences of mobility.

## 5. Conclusions

Very little attention has been paid to the unique mobility needs and experiences of OAEH despite the growing rates of older adult homelessness. OAEH sit in-between the homelessness response, healthcare, and aging services systems, and they often find themselves navigating these disconnected systems via public transit. Attainment of housing and improved health and well-being is contingent on access to reliable transportation, but this condition is not reliably met in many communities. In this study, we have provided a contemporary glimpse into the everyday experience of OAEH as they navigate their communities via public transit, and we have revealed the complex interplay of determinants that influence their mobility in all its forms. For the participants in our study, the physical demands of accessing public transit combined with the cognitive load of interpreting multi-step directions in specific time schedules were often insurmountable. The public transit system was often hostile, such that participants were stigmatized and segregated.

As communities across the country—and globe—struggle to address the evolving needs of OAEH, the findings from this study can inform the work of professionals who work with OAEH and policymakers tasked with urban planning. Professionals must consider the capacity of their clients to successfully navigate public transit, and policymakers need to account for the distinct needs of subpopulations who rely on public transportation. Establishing universal goals such as healthy public transit can serve as a framework within which specific strategies to achieve this goal are tailored to the needs of different groups, such as OAEH. Policymakers should remember that the design of our physical environment, including public transit systems, influences the inclusion and exclusion of marginalized groups and can either inhibit or promote both physical and social mobility.

## Figures and Tables

**Figure 1 ijerph-22-00654-f001:**
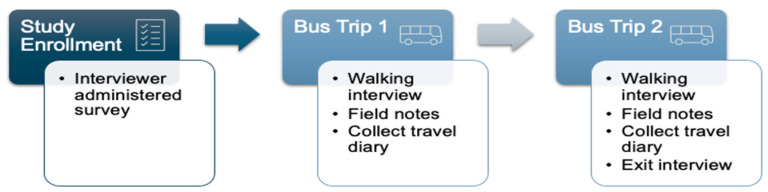
Data collection strategies and timepoints.

**Table 1 ijerph-22-00654-t001:** Participant characteristics.

Demographics		
Age in years, *M* (*SD*)		54.8 (2.9)
Gender, *n* (%)	Male	8 (67)
	Female	4 (33)
Race/ethnicity, *n* (%) *	White	5 (41.2)
	Hispanic	5 (41.2)
	Black/African American	4 (33)
	Native American	1 (8.3)
Chronic conditions, *n* (%)	Hypertension	7 (58.3)
	Diabetes mellitus	4 (33)
	Arthritis/musculoskeletal conditions	4 (33)
	Anxiety/depression	4 (33)
	Bipolar disorder	3 (25)
	Asthma/COPD	3 (25)
	Stroke	1 (8.3)
	High cholesterol	1 (8.3)
	Hepatitis C	1 (8.3)
	HIV	1 (8.3)
	Multiple chronic conditions (≥2)	12 (100)
Physical function, *n* (%)	Within normal limits	0
	Mild impairment	6 (50)
	Moderate impairment	4 (33)
	Severe impairment	2 (17)
Difficulty with transport, *n* (%)	Mild difficulty	1 (8)
	Moderate difficulty	10 (83)
	Extreme difficulty	1 (8)

Note. * Percentages do not equal 100% because participants claimed multiple racial/ethnic identities.

**Table 2 ijerph-22-00654-t002:** Transit activities and planned destinations.

Participant	Transit Activities and Planned Destinations	
	Observation #1	Observation #2
1	Visiting friends and former campsites	Wal-Mart to check if participant had left their ID there when collecting money that his brother had sent via Western Union
2	No particular destination; rode and observed participant’s most frequent bus route	Visiting friends who had recently moved into an apartment
3	A scheduled eye doctor appointment (unsuccessful in making the appointment)	Wal-Mart to see if participant could get his eyes checked via walk-in appointment (unsuccessful); post office to fill out a change of address form
4	Visiting friends and former campsite	The grocery store
5	Homeless services day center to collect mail and personal hygiene products and to select an outfit from the clothing closet	Different homeless navigation center to get a rolling suitcase and inquire about reading glasses, prescription medications, and inhalers
6	Unsuccessful transit trip to homeless navigation center to get an ID due to participant’s inability to walk the distance to the bus stop	N/A
7	T-Mobile store to pay phone bill and Wal-Mart to grocery shop	No particular destination; rode and observed participant’s most frequent bus route
8	A healthcare clinic to schedule appointments	A nearby motel to apply for a housekeeping position.
9	No particular destination; rode and observed participant’s most frequent bus route	Initial destination was a mental health clinic to schedule an appointment, but plans changed when a different bus arrived at the bus stop first; several hotels in city center to inquire about job applications
10	From homeless navigation center to participant’s current campsite	Visiting friends at downtown park
11	Storage facility to get a folder of paperwork	Department of Public Safety for an appointment to get ID renewed
12	A scheduled medical appointment	Homeless navigation center to check mail

Note. Observation #1 indicates the first transit trip observation. Observation #2 indicates the second transit trip observation. N/A indicates that participant 6 was exited from the study after the first observation day due to an inability to independently navigate public transit.

## Data Availability

The original contributions presented in this study are included in the article. Further inquiries can be directed to the corresponding author(s).

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
