# Peer review of "A Community-Engaged Ethnographic Investigation into Public Transit Among Older Adults Experiencing Homelessness"

_ijerph, 2025, doi:10.3390/ijerph22040654_

Round 1
Reviewer 1 Report
Comments and Suggestions for Authors
This study focuses on how homeless older adults use the public transport system to meet their health and social service needs. Using a community-engaged, rapid ethnographic methodology, the article explores the challenges and needs these populations face when utilizing public transport, providing new perspectives and methods for examining accessibility to transport and social services, as well as suggesting practical policy improvements.
However, there are some shortcomings in the following areas, which, if improved, would lead to an improvement in the quality of the article.
- the abstract section succinctly presents the background, methodology and conclusions of the study. It is recommended that a data supplement be included in the background introduction, for example, by including a specific percentage of the homeless population over the age of 50 (lines 9-10).
- The introductionsection describes the current situation of homeless older people, which is problem-aware and more innovative at the level of group specificity. However, the article mentions the idea that existing research has paid little attention to the mobility needs of older people experiencing homelessness (lines 40-42). The number of supporting papers for this idea is weak suggesting more references. It is also possible to refine the gaps in the existing literature at the intersection of ‘older homelessness’ and ‘public transport’.
- 1.1 Sensitizing Framework in the Introduction section could include a graphical visualization of the framework.
- In the Materials and Methods section, the specific modalities of the walking interviews could be described in more detail, such as the length of the interviews and the corresponding frequency of note-taking (lines 164-170).
- In the results section, the travel diary is cited in several places (lines 255-257), and an analysis of its underlying causes can be added to the categorization.
- discussion section, limitations in which the potential interference of the researcher in the interviews with the participants' behaviour is not analyzed (lines 467-484).
- in the conclusion section, the argument is too brief and lacks elaboration on the outlook for future research and policy implications. The value of this study is not adequately highlighted (lines 467-484)
Author Response
This study focuses on how homeless older adults use the public transport system to meet their health and social service needs. Using a community-engaged, rapid ethnographic methodology, the article explores the challenges and needs these populations face when utilizing public transport, providing new perspectives and methods for examining accessibility to transport and social services, as well as suggesting practical policy improvements.
However, there are some shortcomings in the following areas, which, if improved, would lead to an improvement in the quality of the article.
- the abstract section succinctly presents the background, methodology and conclusions of the study. It is recommended that a data supplement be included in the background introduction, for example, by including a specific percentage of the homeless population over the age of 50 (lines 9-10).
Response: Thank you for this suggestion. We have added the percentage of older homeless adults in the abstract. See line one in the abstract.
- The introduction section describes the current situation of homeless older people, which is problem-aware and more innovative at the level of group specificity. However, the article mentions the idea that existing research has paid little attention to the mobility needs of older people experiencing homelessness (lines 40-42). The number of supporting papers for this idea is weak suggesting more references. It is also possible to refine the gaps in the existing literature at the intersection of ‘older homelessness’ and ‘public transport’.
Response: Thank you for this comment. We have revised the introduction section to more clearly highlight and refine the gaps in the existing literature related to older adult homelessness, mobility, and public transport. We removed one sentence from the first paragraph in the introduction (p.3). We have clarified the gaps are both methodological and population-specific. That is, the use of rapid ethnographic assessments to capture a more contextual understanding helps to deepen our understanding of mobility barriers and transportation disadvantage among older people experiencing homelessness in particular. We expanded the fifth paragraph of the introduction (p.4).
- 1.1 Sensitizing Framework in the Introduction section could include a graphical visualization of the framework.
Response: We did not develop a graphical visualization of the framework when we analyzing our data, so we do not have a useful visualization.
- In the Materials and Methods section, the specific modalities of the walking interviews could be described in more detail, such as the length of the interviews and the corresponding frequency of note-taking (lines 164-170).
Response: We have added these suggested details. See second paragraph of data collection subsection (p.7-8)
- In the results section, the travel diary is cited in several places (lines 255-257), and an analysis of its underlying causes can be added to the categorization.
Response: Thank you for this suggestion. We clarified in the analysis section that the travel diaries were analyzed in the same manner as the transcripts and field notes (p.9).
- discussion section, limitations in which the potential interference of the researcher in the interviews with the participants' behaviour is not analyzed (lines 467-484).
Response: This is an excellent point, and we have added language regarding this limitation (p18-19).
- in the conclusion section, the argument is too brief and lacks elaboration on the outlook for future research and policy implications. The value of this study is not adequately highlighted (lines 467-484)
Response: We have expanded the conclusion to further elaborate the contributions of this study. Thank you for this feedback (p.19).
Reviewer 2 Report
Comments and Suggestions for Authors
Initially, the manuscript informs that ‘adults over the age of 50 are the fastest growing sub-population of people experiencing homelessness in the U.S. and have unique mobility needs and transportation barriers.’ The relative originality of the theme lies in the relationships between older people experiencing homelessness (OPEH) and mobility and public transportation, addressing a specific gap in the field and adding information to the subject area compared with other published material. Due to the demand for detailed research, its thematic relevance may be an aspect of interest for the journal's readers.
Thus, the objective was ‘to explore practices, needs, perceptions, and values of unhoused older adults as they relate to accessing and navigating health and social services via public transit.’ Despite the possibility of interpreting the main question addressed by the research, there is no clear specification of it in the article's writing, nor of an investigative hypothesis, items that could already be exposed in the abstract itself. In this context, it is suggested that the keywords do not repeat terms used in the manuscript title to expand the ways of indexing.
Section 1 (Introduction) presents aspects of mobility and public transportation considered essential for older adult quality of life, as well as the growth of PEH, providing data on the U.S. situation and contributions from other studied cases. Subsection 1.1 (Sensitizing Framework) indicates the conceptual structure for assessing unmet travel needs. This introductory section is supported by almost 50% of references from the last decade, with less than 30% relating to the past five years.
In Section 2 (Materials and Methods), it is indicated that the research is guided by principles of community-based participatory research (CBPR), adopted by a workgroup that includes direct service providers, policymakers, researchers, and people with lived experience of homelessness. Although it is widely used, the option of rapid ethnography reveals limitations due to the small sample size (n = 12), with 23 participant observations over 65 hours, field notes, walk-along interviews, and travel diaries, with recruitment (individuals aged at least 50 years and currently experiencing homelessness or housed in a rapid rehousing program) at an unnamed emergency agency. Thus, the possibilities of generalizing the achieved results are reduced.
Informed consent terms were provided and ethical approval was obtained from the University Institutional Review Board, but it is suggested that the respective protocol be specified. The authors ensure that strategies were used to guarantee scientific rigor, dedicating a specific subsection to the subject. Even so, there is the possibility of more detailed methodological information to improve the reproducibility of the research, recommending, among other things, the graphic diagramming of the investigative steps for a proper understanding of the general structure of the methods and techniques applied. In this methodological section, most of the references (more than 85%) are over 10 years old.
Section 3 (Results) presents descriptions, data, and context, with information on the demographic and health-related characteristics of the only twelve individuals enrolled in the study and their transit activities and planned destinations for each observation. Two sub-themes (waiting – Subsection 3.1 – and friction – Subsection 3.2) illustrate the actions and speeches, including some personal testimonies, that constituted the participants' mobility experiences.
In Section 4 (Discussion), the authors argue that obtaining housing and improving the health and well-being of this population depends on access to reliable transportation and state that this requirement is not met in many communities. They also add that ‘the physical demands of accessing public transit combined with the cognitive load of interpreting multi-step directions on specific time schedules was often insurmountable. Further, the public transit system was often hostile which served to stigmatize and segregate participants.’
This last section has all its new references within the last decade, with about 78% relating to the past five years. Of significant scientific importance, Subsection 4.1 reports the study's limitations, such as the small sample size and setting, ‘not generalizable to other OPEH in other geographical locations,’ the possibility of selecting a subset of subjects with a greater ability to participate, and the lack of verification of the authors' interpretations with the participants. They also indicate that ‘future research should investigate mobility and transportation among rural OPEH’, as well as ’the impact of different diagnoses and functional abilities on participants’ experiences of mobility.’
The conclusions (Section 5) are very concise, stating that ‘OPEH sit in-between the homelessness response, healthcare, and aging services systems, and they often find themselves navigating these disconnected systems via public transit.’ In summary, they could be more consistent with the evidence and arguments presented and provide answers to a main question posed and/or results of hypothesis testing
In general terms, the references are appropriate, with occasional data gaps. Overall, more than 55% correspond to the last 10 years, with about 35% relating to the last five years. Thus, they could be more up-to-date, especially in the topics previously mentioned. The manuscript contains only two tables, which are indispensable for understanding the content, but it lacks graphical information, such as methodological diagrams, and other information already mentioned, including the study location (metropolitan area of the south-central U.S.) and its regional characteristics, among others that may influence the configuration of the results.
Author Response
Initially, the manuscript informs that ‘adults over the age of 50 are the fastest growing sub-population of people experiencing homelessness in the U.S. and have unique mobility needs and transportation barriers.’ The relative originality of the theme lies in the relationships between older people experiencing homelessness (OPEH) and mobility and public transportation, addressing a specific gap in the field and adding information to the subject area compared with other published material. Due to the demand for detailed research, its thematic relevance may be an aspect of interest for the journal's readers.
Response: We are glad that this topic and the introduction is of interest to the reviewer, and we agree that it is relevant for the journal’s readers. We appreciate the reviewer’s helpful suggestions for improvement.
Thus, the objective was ‘to explore practices, needs, perceptions, and values of unhoused older adults as they relate to accessing and navigating health and social services via public transit.’ Despite the possibility of interpreting the main question addressed by the research, there is no clear specification of it in the article's writing, nor of an investigative hypothesis, items that could already be exposed in the abstract itself. In this context, it is suggested that the keywords do not repeat terms used in the manuscript title to expand the ways of indexing.
Response: We added some context to the discussion in order to clarify how the results address the main study objective (p.16). We do not have an investigative hypothesis because the qualitative methods used in this study do not call for a hypothesis. We have revised some of the keywords per the reviewer’s suggestion (p.2).
Section 1 (Introduction) presents aspects of mobility and public transportation considered essential for older adult quality of life, as well as the growth of PEH, providing data on the U.S. situation and contributions from other studied cases. Subsection 1.1 (Sensitizing Framework) indicates the conceptual structure for assessing unmet travel needs. This introductory section is supported by almost 50% of references from the last decade, with less than 30% relating to the past five years.
In Section 2 (Materials and Methods), it is indicated that the research is guided by principles of community-based participatory research (CBPR), adopted by a workgroup that includes direct service providers, policymakers, researchers, and people with lived experience of homelessness. Although it is widely used, the option of rapid ethnography reveals limitations due to the small sample size (n = 12), with 23 participant observations over 65 hours, field notes, walk-along interviews, and travel diaries, with recruitment (individuals aged at least 50 years and currently experiencing homelessness or housed in a rapid rehousing program) at an unnamed emergency agency. Thus, the possibilities of generalizing the achieved results are reduced.
Response: We agree that the results of this study are not able to be widely generalizable. However, generalizability is not the intent of qualitative research. Instead, we have addressed transferability in several ways including providing detailed descriptions of the participants, methods, data, and context so that readers can make connections between elements of this study and their own experience (p.9-10).
Informed consent terms were provided and ethical approval was obtained from the University Institutional Review Board, but it is suggested that the respective protocol be specified. The authors ensure that strategies were used to guarantee scientific rigor, dedicating a specific subsection to the subject. Even so, there is the possibility of more detailed methodological information to improve the reproducibility of the research, recommending, among other things, the graphic diagramming of the investigative steps for a proper understanding of the general structure of the methods and techniques applied. In this methodological section, most of the references (more than 85%) are over 10 years old.
Response: The IRB protocol number has been added (p.7). We used references that are generally considered to be gold standard for our methods, thus we did not focus on the publication dates for these references. We have added a representative timeline in graphical format to aid in understanding the participant experience in the study and the methods used (p.8).
Section 3 (Results) presents descriptions, data, and context, with information on the demographic and health-related characteristics of the only twelve individuals enrolled in the study and their transit activities and planned destinations for each observation. Two sub-themes (waiting – Subsection 3.1 – and friction – Subsection 3.2) illustrate the actions and speeches, including some personal testimonies, that constituted the participants' mobility experiences. In Section 4 (Discussion), the authors argue that obtaining housing and improving the health and well-being of this population depends on access to reliable transportation and state that this requirement is not met in many communities. They also add that ‘the physical demands of accessing public transit combined with the cognitive load of interpreting multi-step directions on specific time schedules was often insurmountable. Further, the public transit system was often hostile which served to stigmatize and segregate participants.’
This last section has all its new references within the last decade, with about 78% relating to the past five years. Of significant scientific importance, Subsection 4.1 reports the study's limitations, such as the small sample size and setting, ‘not generalizable to other OPEH in other geographical locations,’ the possibility of selecting a subset of subjects with a greater ability to participate, and the lack of verification of the authors' interpretations with the participants. They also indicate that ‘future research should investigate mobility and transportation among rural OPEH’, as well as ’the impact of different diagnoses and functional abilities on participants’ experiences of mobility.’
The conclusions (Section 5) are very concise, stating that ‘OPEH sit in-between the homelessness response, healthcare, and aging services systems, and they often find themselves navigating these disconnected systems via public transit.’ In summary, they could be more consistent with the evidence and arguments presented and provide answers to a main question posed and/or results of hypothesis testing
Response: We revised and extended the discussion and conclusion sections in order to provide clearer answers to our main research objective (p.16-19).
In general terms, the references are appropriate, with occasional data gaps. Overall, more than 55% correspond to the last 10 years, with about 35% relating to the last five years. Thus, they could be more up-to-date, especially in the topics previously mentioned. The manuscript contains only two tables, which are indispensable for understanding the content, but it lacks graphical information, such as methodological diagrams, and other information already mentioned, including the study location (metropolitan area of the south-central U.S.) and its regional characteristics, among others that may influence the configuration of the results.
Response: We added a study setting sub-section in the methods section in order to add detail and context regarding regional characteristics (p.6). We also added a graphic timeline as a methodological diagram (p.8).
Reviewer 3 Report
Comments and Suggestions for Authors
Dear authors,
The article is well structured, following a logical flow from introduction to methods, results and conclusions.
The article "A Community-Engaged Ethnographic Investigation into Public Transit Among Older Adults Experiencing Homelessness" explores the mobility experiences of older persons experiencing homelessness (OPEH) in the context of public transportation use. The study employs a rapid ethnographic methodology to highlight barriers and challenges. The article addresses a significant social and public health issue, filling a gap in the specialized literature regarding the mobility of older homeless persons.
The study is based on a small sample (12 participants) from a single metropolitan area, which limits the generalizability of the results. A more extensive discussion about how these limitations affect the external validity of the study is needed.
The article does not explicitly formulate testable hypotheses, which could improve clarity and scientific rigor. I suggest that the authors formulate clear hypotheses in future studies.
Although the authors mention using multiple data sources, it is not clear how these were triangulated to strengthen internal validity. I suggest a more detailed discussion about how the different data sources were integrated and validated.
The article provides valuable practical recommendations for policymakers, urban planners, and social service providers, highlighting the need for specific interventions to improve OPEH mobility.
The results have significant implications for the field of public health and gerontology, contributing to a deeper understanding of the needs of this vulnerable population.The study could explore more deeply how the results could inform public policy and urban planning, offering concrete suggestions for improving public transportation systems.
The discussion of the study's implications could be expanded to include a more detailed analysis of how transportation barriers affect access to health and social services. It might also be useful to discuss how these barriers can be overcome through specific interventions.
The study is well conducted and makes valuable contributions to the understanding mobility experiences of older persons experiencing homelessness.
Good luck!
Author Response
The article is well structured, following a logical flow from introduction to methods, results and conclusions.
The article "A Community-Engaged Ethnographic Investigation into Public Transit Among Older Adults Experiencing Homelessness" explores the mobility experiences of older persons experiencing homelessness (OPEH) in the context of public transportation use. The study employs a rapid ethnographic methodology to highlight barriers and challenges. The article addresses a significant social and public health issue, filling a gap in the specialized literature regarding the mobility of older homeless persons.
The study is based on a small sample (12 participants) from a single metropolitan area, which limits the generalizability of the results. A more extensive discussion about how these limitations affect the external validity of the study is needed.
Response: Thank you for this feedback. We agree that the results of this study are not able to be widely generalizable. However, generalizability is not the intent of qualitative research. Instead, we have addressed transferability in several ways including providing detailed descriptions of the participants, methods, data, and context so that readers can make connections between elements of this study and their own experience. We have extended the discussion in the limitations section to more clearly address this concern (p.18-19).
The article does not explicitly formulate testable hypotheses, which could improve clarity and scientific rigor. I suggest that the authors formulate clear hypotheses in future studies.
Response: Thank you for this feedback. We do not typically use hypotheses in qualitative research such as that presented in the current study. We will keep this in mind for future work that is focused on hypothesis testing.
Although the authors mention using multiple data sources, it is not clear how these were triangulated to strengthen internal validity. I suggest a more detailed discussion about how the different data sources were integrated and validated.
Response: Thank you for this suggestion. We have made some edits to further elucidate how the integration and validation of data contributed to internal validity (p.9).
The article provides valuable practical recommendations for policymakers, urban planners, and social service providers, highlighting the need for specific interventions to improve OPEH mobility.
Response: Thank you for this positive feedback.
The results have significant implications for the field of public health and gerontology, contributing to a deeper understanding of the needs of this vulnerable population. The study could explore more deeply how the results could inform public policy and urban planning, offering concrete suggestions for improving public transportation systems. The discussion of the study's implications could be expanded to include a more detailed analysis of how transportation barriers affect access to health and social services. It might also be useful to discuss how these barriers can be overcome through specific interventions.
Response: Thank you for this suggestion. We have expanded the discussion section to include concrete suggestions for planners and improvements to public transit systems and have also offered suggestions for individual- and client-focused interventions to improve and aid in transportation accessibility (p.16-18).
The study is well conducted and makes valuable contributions to the understanding mobility experiences of older persons experiencing homelessness.
Response: Thank you for this positive feedback.
Good luck!
Round 2
Reviewer 2 Report
Comments and Suggestions for Authors
Preliminarily, we reiterate our opinion regarding the thematic relevance of the manuscript and its relative originality, addressing a specific gap in the field and adding information to the subject area, fostering possible interest for the journal's readers.
As expected in quality scientific works, there must be clarity of its outlines. We agree with the assertion of the possibility, in certain studies, of the absence of a hypothesis to be tested, and this was merely a suggestion to improve the understanding of the real purposes of the research due to the insufficient specification of the main question to be addressed. In this new version, the investigative objective itself seems less evident than in the previous one. On the other hand, we believe that the change in keywords is adequate, albeit partial, in order to expand indexing possibilities.
We also agree with the authors' response that the results of this study are not able to be widely generalizable. However, this would be a desirable aspect, even in the case of qualitative research. Anyway, the explanations of the study's limitations have been expanded in this version, detailing the transferability of information in several ways.
The old references related to the methods have been justified, notwithstanding the possibility of their updating in light of the evolutionary role of science itself. The added representative timeline in graphical format aids in understanding the participant experience in the study, but not the methods used. However, the additions to the text compensate for this deficiency.
Although not mandatory in certain manuscripts, the conclusion section is still considered very succinct and there remains a need to review the references. It is worth noting failures in page specifications in the authors' responses regarding the new version of the manuscript, which may have caused difficulties in interpretation during this final review.
Author Response
Comment 1: Preliminarily, we reiterate our opinion regarding the thematic relevance of the manuscript and its relative originality, addressing a specific gap in the field and adding information to the subject area, fostering possible interest for the journal's readers.
Response: Thank you for this feedback. We appreciate the general positive feedback from the review. Some of the comments are difficult for us to interpret in terms of what is being asked of us. We have noted below where we have made some edits and where we are not sure if there is something for us to revise.
Comment 2: As expected in quality scientific works, there must be clarity of its outlines. We agree with the assertion of the possibility, in certain studies, of the absence of a hypothesis to be tested, and this was merely a suggestion to improve the understanding of the real purposes of the research due to the insufficient specification of the main question to be addressed. In this new version, the investigative objective itself seems less evident than in the previous one. On the other hand, we believe that the change in keywords is adequate, albeit partial, in order to expand indexing possibilities.
Response: We are not quite sure how to interpret this comment in terms or if there is something that we need to revise.
Comment 3: We also agree with the authors' response that the results of this study are not able to be widely generalizable. However, this would be a desirable aspect, even in the case of qualitative research. Anyway, the explanations of the study's limitations have been expanded in this version, detailing the transferability of information in several ways.
Response: We are glad that the expanded explanation of the study's limitations and the expanded discussion on transferability have clarified this issue.
Comment 4: The old references related to the methods have been justified, notwithstanding the possibility of their updating in light of the evolutionary role of science itself. The added representative timeline in graphical format aids in understanding the participant experience in the study, but not the methods used. However, the additions to the text compensate for this deficiency.
Response: We are glad that the graphical timeline and additions to the text have clarified the methods.
Although not mandatory in certain manuscripts, the conclusion section is still considered very succinct and there remains a need to review the references. It is worth noting failures in page specifications in the authors' responses regarding the new version of the manuscript, which may have caused difficulties in interpretation during this final review.
Response: We are also not sure how to interpret this comment. It was unclear from the previous review if the reviewer was asking for updated references, as the comments were just statements of the percentages of references from different time periods. To err on the side of caution, we have updated a few of the references where it was possible to do so. We are also not sure about the comment regarding failures of page specifications. We did include page numbers on the response to reviewers, but perhaps the version that is available for the reviewers had a different format which caused a misalignment in page numbers. We have also extended the conclusion per the reviewer’s suggestion.